# Removal of Cd(II) from Micro-Polluted Water by Magnetic Core-Shell Fe_3_O_4_@Prussian Blue

**DOI:** 10.3390/molecules26092497

**Published:** 2021-04-25

**Authors:** Xinxin Long, Huanyu Chen, Tijun Huang, Yajing Zhang, Yifeng Lu, Jihua Tan, Rongzhi Chen

**Affiliations:** 1College of Resources and Environment, University of Chinese Academy of Sciences, Huaibei Town 380, Huairou District, Beijing 101408, China; longxinxin16@mails.ucas.ac.cn (X.L.); chy1208@foxmail.com (H.C.); tanjh@ucas.ac.cn (J.T.); 2Key Laboratory of Groundwater Circulation and Evolution, School of Water Resources and Environment, China University of Geosciences, No. 29 Xueyuan Road, Haidian District, Beijing 100083, China; 3School of Ecology and Environmental Science, Yunnan University, Kunming 650091, China; huangtijun2021@163.com (T.H.); lyftoday@126.com (Y.L.); 4Sino-Japan Friendship Centre for Environmental Protection, Beijing 100029, China; zhangyajing@edcmep.org.cn; 5State Key Laboratory of Organic Geochemistry and Guangdong Key Laboratory of Environmental Protection and Resources Utilization, Guangzhou Institute of Geochemistry, Chinese Academy of Sciences, Guangzhou 510640, China

**Keywords:** adsorption, Prussian blue, Fe_3_O_4_@PB, cadmium removal, micro-polluted water

## Abstract

A novel core-shell magnetic Prussian blue-coated Fe_3_O_4_ composites (Fe_3_O_4_@PB) were designed and synthesized by in-situ replication and controlled etching of iron oxide (Fe_3_O_4_) to eliminate Cd (II) from micro-polluted water. The core-shell structure was confirmed by TEM, and the composites were characterized by XRD and FTIR. The pore diameter distribution from BET measurement revealed the micropore-dominated structure of Fe_3_O_4_@PB. The effects of adsorbents dosage, pH, and co-existing ions were investigated. Batch results revealed that the Cd (II) adsorption was very fast initially and reached equilibrium after 4 h. A pH of 6 was favorable for Cd (II) adsorption on Fe_3_O_4_@PB. The adsorption rate reached 98.78% at an initial Cd (II) concentration of 100 μg/L. The adsorption kinetics indicated that the pseudo-first-order and Elovich models could best describe the Cd (II) adsorption onto Fe_3_O_4_@PB, indicating that the sorption of Cd (II) ions on the binding sites of Fe_3_O_4_@PB was the main rate-limiting step of adsorption. The adsorption isotherm well fitted the Freundlich model with a maximum capacity of 9.25 mg·g^−1^ of Cd (II). The adsorption of Cd (II) on the Fe_3_O_4_@PB was affected by co-existing ions, including Cu (II), Ni (II), and Zn (II), due to the competitive effect of the co-adsorption of Cd (II) with other co-existing ions.

## 1. Introduction

Water pollution caused by heavy metals is of great concern due to their bioaccumulation, non-biodegradation, and high toxicity [1,2]. Cadmium (Cd) is extracted from zinc ore or sulfur cadmium ore as a by-product, which is widely used in paints, batteries (like nickel-cadmium batteries), pigments, and electroplating [3,4]. A growing body of evidence shows that long-term exposure to cadmium may cause adverse effects on human health [5]. The maximum carcinogenic risk of Cd was suggested at the level of 10^−7^ μg/L for the individual through different exposure pathways [6]. In the past few years, drinking water resources that have been subjected to heavy metal micro-pollution, such as Cd micro-polluted water, have generated various concerns. Chakrabarty and Sharma reported that higher levels of Cd (an average of 25 μg/L) in wells were caused by geogenic contamination in Assam, India [7]. Concentrations of Cd were found to be in the range of 7.1~12.3 μg/L in the rivers around Dhaka, Bangladesh [8]. Stricter legislation on pollution emissions and concentrations in the environment has been enforced. Cadmium has been classified as a carcinogen (Group IA) by the International Agency for Research on Cancer (IARC). The US Environmental Protection Agency (EPA) has set the maximum level of cadmium to be less than 5 μg/L for drinking water [9]. As one of the listed high-priority pollutants, the Drinking Water Regulation Limit (DWRL) for Cd is also 5 μg/L [10,11]. However, as micro-polluted water treatment has mainly focused on low concentrations of nitrogen and organic pollutants in natural environments, less attention has been paid to removing heavy metals such as Cd (II) from drinking water. Since natural water bodies contain Cd (II) pollution at relatively low concentrations, the treatment of Cd (II)-polluted water should be given special consideration as micro-polluted water. Developing efficient water purification procedures for the removal of Cd from micro-polluted water is extremely urgent.

Various conventional technologies such as ion-exchange [12], adsorption [13,14], membrane filtration [15,16], chemical precipitation [17] and biological remediation [18] have been applied for Cd (II) removal from wastewater. Among these methods, the adsorption technique is an attractive approach for water treatment, especially if the adsorbent is low in cost and easy to operate, separate, and regenerate [19]. The choice of adsorbents is one of the most critical steps in the successful application of adsorption-based treatment techniques. Several materials have been previously evaluated for the removal of Cd (II) from contaminated water [20], including natural materials such as zeolites, clay [21], diatomite [22], bean-coat [23], activated carbon [24], biochar [25], mesoporous aluminosilicates [26], and nanomaterials such as nanochitosan [27], TiO_2_ nanotubes [28], carbon-based 3D architectures [29], and Metal-organic frameworks (MOFs) [30]. However, these adsorbents are difficult to separate and recover from aqueous solutions. The commonly used recovery methods as filtration and centrifugation are time-consuming and expensive, which limits their reuse and may cause secondary pollution. Meanwhile, in studies where high initial concentrations of target pollutants were applied to evaluate the properties of adsorbents, the residual Cd concentration in solutions after treatment presents potential risks in drinking water and may exceed the DWRLs.

There is an urgent need to develop an economical and highly efficient adsorbent that can be easily prepared and separated from a solution to remove Cd from micro-polluted water. Magnetic nanoparticles that can be separated from an external magnetic field have attracted increasing research attention [31]. Unmodified Fe_3_O_4_ magnetic nanoparticles have been used to remove and separate heavy metals from wastewater [32,33]. The superparamagnetic composite materials prepared by surface modification of Fe_3_O_4_ magnetic nanoparticles have been reported to have many applications. As a face-centered cubic lattice, Prussian blue (PB) has attracted significant attention from both theoretical and applied scientists due to its unique properties and wide applications [34]. Scientists proposed the use of magnetic Prussian blue (MPB) composites to remove toxic ions from polluted water. Sabaki et al. used the precipitation method to synthesize PB-Fe_3_O_4_, which could play a vital role in removing radioactive metal ions (Cs^+^) from aqueous solutions and was found to maintain its adsorptive capacity in high ionic strength NaCl salt solution [35]. Thammawong et al. reported the development of magnetic Prussian blue nano sorbent with high sorption capacity for Cs^+^ [36]. Uogintė et al. utilized magnetic Prussian blue nano sorbent (MPB) for the removal of Cu (II), Co (II), Ni (II), and Pb (II) from aqueous solutions [37]. Results showed that MPB was suitable for the removal processes and retained a high sorption capacity. While several authors have described the use of MPB for the sorption of radionuclides from contaminated solutions, there has been little research on the use of MPB for removing Cd (II) micro-polluted water.

The adsorption capacity of several adsorbents has been proved to be significantly improved by surface modification [38,39,40,41]. Inspired by this, this study aimed to synthesize core-shell MPB spheres (Fe_3_O_4_@PB) to be used as adsorbent materials to remove Cd (II) species from micro-polluted water. Fe_3_O_4_@PB was characterized by various methods. The effects of reaction time, dosage, co-existing ions, pH value, and initial concentration of solutions on the adsorption of Cd (II) onto Fe_3_O_4_@PB were explored. The possible interaction mechanisms between Cd (II) on Fe_3_O_4_@PB were analyzed by FT-IR and Zeta potential, and the adsorption isotherm and kinetics were also employed to discuss the adsorption mechanisms.

## 2. Results and Discussion

### 2.1. Characterization of Fe_3_O_4_@PB

The micrographic features of Fe_3_O_4_@PB were characterized by scanning electron microscopy (SEM) and transmission electron microscopy (TEM). The exhibited SEM images of Fe_3_O_4_@PB in Figure 1a,b show a uniform spherical shape with a particle diameter of about 150 nm. The structures of Fe_3_O_4_ and Fe_3_O_4_@PB were further observed by TEM, as shown in Figure 1c,d, respectively. The Fe_3_O_4_ nanoparticles are monodispersed with a smooth surface. As exhibited in Figure 1d, Fe_3_O_4_ is coated by a rough layer of agglomerated PB particles, meaning that PB nanoparticles successfully adhered to the surface of Fe_3_O_4_. Figure 1e verifies the typical core-shell structure.

The X-ray diffraction (XRD) patterns of the as-synthesized adsorbents were recorded in Figure 2a, which determined the crystalline structure and main phase of the composites. Similar peaks of Fe_3_O_4_ and Fe_3_O_4_@PB at 2*θ* 18.29, 30.14, 35.49, 37.14, 43.13, 53.52, 57.04, and 62.64° were indexed as (111), (220), (311), (222), (400), (422), (511), and (440) planes, respectively, well-matched with the previous researches [42,43]. Besides, Fe_3_O_4_@PB displays four predominant 2θ peaks compared with Fe_3_O_4_ in the range of 17–40° corresponding to the Bragg planes of (200), (220), (400), and (420) [44], which belong to the face-centered cubic lattice structure of PB. These prove that the composites are designed by the synergistic effect between PB shell and Fe_3_O_4_ core, and the introduction of Fe_3_O_4_ did not affect the crystallinity of PB nanoparticles.

As presented in Figure 2b, FTIR was further employed to compare the functional structure of Fe_3_O_4_ and Fe_3_O_4_@PB. The peak of Fe−O locates at 594 cm^−1^ for Fe_3_O_4_ [45] and shows a slight blue shift for core-shell Fe_3_O_4_@PB. The same typical bands revealed by FTIR indicate the introduction of Fe_3_O_4_ in the core-shell structure, which coincides with the result from XRD. The major peak in the vicinity of 2092 cm^−1^ assigned to the −CN stretching vibration [46] and the stretching bands at 599 cm^−1^ and 501 cm^−1^ related to the formation of Fe−CN−Fe [47], demonstrating the occurrence of Fe_3_O_4_@PB. The coordination between the nitrogen lone electron pair of −CN in the PB and the 3p^3^ hybrid orbital formed by the Cd (II) 5s and 5p orbitals may have contributed to the differences shown by FTIR after adsorption. Previous studies determined that O−H groups bind to iron cations on the surface of Fe_3_O_4_ in aqueous water [48,49]. However, the stretching band near 3300 cm^−1^ revealed that the O−H stretching vibration was very weak, which can be attributed to vacuum desiccation in the synthesis process, making the spectrum band of O−H groups on the particle surface around 3300 cm^−1^ nearly unobservable. The band at 1654 cm^−1^ could be attributed to the bending vibration of O−H, indicating the existence of interstitial water in the magnetic PB nanocomposites.

The pore structure of Fe_3_O_4_@PB was characterized by N_2_ adsorption/desorption analysis at 77.3 K and generated by the adsorption branch of the isotherms using the Barrett–Joyner–Halenda (BJH) method. Figure 2c displays the type II adsorption isotherm without an obvious hysteresis loop of Fe_3_O_4_@PB, indicating a micropore-dominated structure. In general, it was difficult for N_2_ molecules to enter the intrinsic micropores inside due to the occupation of water molecules and other ions, which means that there are mesopores on the surface of Fe_3_O_4_@PB [50]. As shown in the corresponding pore size distribution curve, the pore volume of Fe_3_O_4_@PB is mainly contributed by pore size in the range of 1.7–5.5 nm, verifying the confounding of micropores and mesopores.

Magnetic hysteresis (M−H) curves of the Fe_3_O_4_ and Fe_3_O_4_@PB nanoparticles show a typical magnetic hysteresis loop (Figure 2d). The saturation magnetization (M_S_) of Fe_3_O_4_ was 70 emu/g at room temperature. After coated by PB, the saturation magnetization decreased to 12.0 emu/g, which is similar to the M_S_ 12.07 emu·g^−1^ of the magnetic PB composites prepared by Jang et al. [51]. Despite the decreases in saturation magnetization with outer shell PB, the magnetization still could guarantee the recyclability of Fe_3_O_4_@PB sorbents.

The Zeta potentials of the Fe_3_O_4_@PB at varied pH at room temperature were collected by SOE-070 nanoparticle size and Zeta potentiometer (Delsa Nano C/Z, Beckman Coulter, Brea, CA, USA) (Figure 3). The point of zero charges (pH_PZC_) of Fe_3_O_4_ is approximately equal to 6.19, indicating it would be positively charged at pH below 6.19 while negatively charged at higher pHs. The Zeta potentials all had negative values within the pH range of 3–11. The negative values dramatically decreased as pH increased from 2 to 6; however, the Zeta potentials values changed only slightly as pH rose from 6 to 11. This could be attributed to the gradual saturation of hydroxyl groups on Fe_3_O_4_@PB surface with the increase in solution pH. Thus, Fe_3_O_4_@PB was found to be negatively charged in the entire environmentally relevant pH range, which is beneficial to the adsorption of positively charged cations. The pH_PZC_ of Fe_3_O_4_@PB is around 2.31, much lower than that of Fe_3_O_4_, implying the stronger electrostatic attraction of Fe_3_O_4_@PB.

### 2.2. Effect of Fe_3_O_4_@PB Dosage

As shown in Figure 4a, the increasing dosage of Fe_3_O_4_@PB from 0.1 to 2 g·L^−1^ significantly improved the removal efficiency of Cd (II). However, the reaction rate remained almost unchanged when the sorbent dosage was further increased to 4 g·L^−1^. Previous studies suggested that more adsorption sites become available for metal uptake with the sorbent dosage increases [52]. At sorbent dosages >1 g·L^−1^, the incremental Cd (II) ion removal slowed as the metal ion concentrations reached equilibrium on the surface. The decrease in *q_e_* may have been due to the generation of unsaturated adsorption sites through the adsorption reaction when dosages increased. Another possible explanation for the decrease in *q_e_* may have been particle interactions such as aggregation due to high dosages of sorbent, which then led to a decrease in the total surface area of the adsorbent [53]. Hence, a dosage of 1–2 g·L^−1^ was determined to be the most suitable for Cd (II) removal in terms of efficiency and cost.

### 2.3. Effect of Initial pH

The pH value of aqueous solutions was one of the vital factors that significantly affected the adsorption of Cd (II) on the water-absorbent interfaces [54], the surface charge of the adsorbents, and the contaminant species [55]. Cadmium species in deionized water include Cd^2+^, Cd(OH)^+^, Cd(OH)_2_^0^, and Cd(OH)_2(s)_ [56]. At pH < 6, Cd^2+^ was the only ionic species present in aqueous solutions, while Cd^2+^ and Cd(OH)^+^ were the dominant cadmium species at pH < 8, and Cd(OH)_2_ precipitation began to form at pH > 8. The pH of solutions influences metal ion adsorption through the competition between metal ions and H^+^ ions for active sorption sites [57]. In order to establish how pH affects Cd (II) ion sorption onto Fe_3_O_4_@PB, batch studies were conducted at different initial pH values in the range of 2 to 11.

As shown in Figure 4b, the adsorption efficiency was only 2.24% at the initial pH of 2; then, it increased from 12.53% to 98.78% as the pH increased from 3 to 6 and leveled off in the pH range of 6–9. The low metal sorption at pH of 2 was likely due to active site protonation, which led to competition between H^+^ and Cd (II) to occupy the adsorption sites [58]. The lower concentration of H^+^ weakened the competition adsorption of H^+^ and Cd (II) ions at a higher pH level [59]. Conversely, Cd (II) ions are prone to the formation of Cd(OH)^+^ and Cd(OH)_2_ at pH > 6 (aggregation effect between Cd (II) with OH^−^), reducing the removal efficiency at high pH values. In alkaline conditions, precipitation plays a major role in the removal of Cd (II) due to the formation of Cd(OH)_2(S)_ precipitate. The precipitation of metal hydroxides into pores or spaces around the adsorbent particles is nearly impossible as the adsorption process is kinetically faster than precipitation. In this study, the maximum removal efficiency (98.78%) of Cd (II) was acquired at a pH of 6 for Fe_3_O_4_@PB. The resulting Cd (II) concentrations after adsorption were far below the 5 μg/L limit designated by the DWRL.

### 2.4. Effect of Coexisting Ions

Figure 5 indicates that the effect of co-existing ions on the adsorption of Cd (II) onto Fe_3_O_4_@PB depended on the variety of ions (Cu (II), Ni (II), and Zn (II)). As the concentration of the co-existing ions rose, competition with other heavy metal ions for the adsorption sites caused Cd (II) removal efficiency to decrease [60]. The hydrated radii of Cd (II), Cu (II), Ni (II), and Zn (II) were 4.26 Å, 4.19 Å, 4.04 Å, and 4.30 Å, respectively. Small differences in hydrated radiuses are a significant cause of lattice competition. Fe_3_O_4_@PB can remove not only Cd (II) but also Cu (II), Ni (II), and Zn (II) from wastewater. Therefore, these new insights provide valuable information for the application of Fe_3_O_4_@PB to remove heavy metal co-contamination from micro-polluted water.

### 2.5. Effect of Contact Time and Adsorption Kinetic

The Cd (II) adsorption on Fe_3_O_4_@PB shows two distinct phases in Figure 6a: a rapid initial phase over the first 2 h and a much slower sorption phase to reach equilibrium within 4 h. The adsorption capacity of Cd (II) on Fe_3_O_4_@PB increased as time passed and then reached a plateaued equilibrium. The remarkable increase in Cd (II) adsorption capacity at the initial step was due to the existence of plentiful active sites on the adsorbent surface. After that, as the majority of active surface sites were occupied by Cd (II), the adsorption process slowed until it reached a plateaued equilibrium due to insufficient remaining binding sites for Fe_3_O_4_@PB to absorb Cd (II).

To investigate the adsorption mechanism during the adsorption process, kinetic models including the pseudo-first-order, pseudo-second-order, intraparticle diffusion, and Elovich models were used to evaluate the experimental data. The parameters results are shown in Table 1. Cd (II) adsorption onto Fe_3_O_4_@PB followed the pseudo-first-order and Elovich models well with the correlation coefficient R^2^ > 0.99. Moreover, the calculated adsorption capacity from the pseudo-first-order model was much closer to the experimental value. The fitness of the pseudo-first-order model indicates a surface reaction controlled process, in which the adsorption of Cd (II) onto Fe_3_O_4_@PB may depend on the surface active sites and the affinity between cadmium ions and the adsorbents [61]. Elovich model has been proved to describe the solid-liquid interaction in chemical adsorption processes effectively [62]. All results above indicate that the sorption of Cd (II) ions on the binding sites of Fe_3_O_4_@PB was the main rate-limiting step of adsorption.

### 2.6. Effect of Initial Concentration and Adsorption Isotherm

The effect of Cd (II) initial concentrations on the adsorption capacity of Fe_3_O_4_@PB is plotted in Figure 6b. The adsorption quantity increased rapidly when the Cd(II) concentration increased from 0 to 1000 μg/L. Then, the adsorption quantity increased slower with the Cd(II) concentration increased further. At lower concentrations of Cd (II), the available sorption sites are ample with high attraction towards Cd (II), the increase in initial concentration drives the adsorption of Cd(II) from aqueous solution onto the adsorbent surface [63], achieving the rapid increase in equilibrium adsorption capacity. As the initial concentration continues to increase, the adsorption sites remain the same while the number of adsorbate molecules increased, leading to the competition of more Cd (II) due to the saturation effect [64].

The isotherm experiments were conducted under optimal aqueous conditions (a duration of 4 h, a pH of 6, and a temperature of 25 °C). The isotherm fitness plots and parameters are shown in insets of Figure 6b and Table 1, respectively. The Langmuir isotherm was based on reaction hypotheses and assumed that monolayer adsorption occurs on the surface sorption part without interaction between adsorbates, while the Freundlich isotherm was used to demonstrate that physicochemical adsorption on heterogeneous surfaces was related to multilayer adsorption with varying affinities [65]. Based on the higher determination coefficient (*R*^2^), Cd (II) removal by Fe_3_O_4_@PB fitted the Freundlich model better than the Langmuir model, indicating that there were several mechanisms of Cd (II) adsorption on the surface of Fe_3_O_4_@PB. Integrating the results of both Zeta potential and FTIR suggested that electrostatic interaction and metal complexation may be the main adsorption mechanisms. The exponent *n* in the Freundlich model was also an indicator for predicting whether an adsorption system is favorable. The 1/n value in this study was 0.4378, which fell into the range of 0.1–1, suggesting that Cd (II) was adsorbed efficiently onto Fe_3_O_4_@PB [66]. The maximum adsorption capacity of Fe_3_O_4_@PB for Cd (II) was estimated as 9.25 mg·g^−1^, which provides new insight into the removal of heavy metal ions from micro-polluted water.

## 3. Materials and Methods

### 3.1. Reagents

Ferric chloride hexahydrate (FeCl_3_·6H_2_O), cadmium chloride (CdCl_2_·5/2H_2_O), potassium chloride (KCl), ferrosoferric oxide (Fe_3_O_4_), potassium ferricyanide (K_3_Fe(CN)_6_), and hydrochloric acid (HCl) were purchased from Macklin and Beijing Chemical Works, China.

### 3.2. Synthesis of Fe_3_O_4_@PB Magnetic Adsorbent

One-pot coordination replication and etching were used to prepare the Fe_3_O_4_@PB as illustrated in Scheme 1. Typically, 160 mg of Fe_3_O_4_, 64 mg of K_3_Fe(CN)_6_, and 895 mg FeCl_3_ were added to 120 mL distilled water in a vial. After ultrasonic radiation and dispersion for 30 min, 1 M of HCl solution (120 mL) was added to the mixture. The suspension was then placed into a vapor bath and shaken under constant temperature conditions (25 °C) for 15 h. Finally, the Fe_3_O_4_@PB nanoparticles were separated from the solution using a permanent magnet and cleaned with ultrapure water 3 times. The product was dried overnight in a vacuum oven at 50 °C.

### 3.3. Characterization and Analytical Methods

The morphology and size of Fe_3_O_4_@PB were observed by scanning electron microscopy (SEM, Hitachi SU-8010, Japan) and high-resolution transmission electron microscopy (TEM, JEOL JEM-2100 F). The presence of PB-modified Fe_3_O_4_ was confirmed by powder X-ray diffraction (XRD, Rigaku Smart Lab, Tokyo, Japan). FTIR spectra of samples were recorded by a Bruker Vertex 70 (Germany) in the wavenumber range of 400–4000 cm^−1^ at a resolution of 4 cm^−1^ using the KBr pellet method. The specific surface area of Fe_3_O_4_@PB was determined by Brunauer-Emmett-Teller (BET, Micrometrics ASAP 2460 instrument, Norcross, GA, USA) method. The magnetic properties were measured by a superconducting quantum interference device (Quantum Design SQUID-VSM, San Diego, CA, USA) in 300 K. The zeta potentials for pH values from 2 to 11 were collected on SOE-070 nanoparticle size and Zeta potentiometer (Delsa Nano C/Z, Beckman Coulter, Brea, CA, USA).

### 3.4. Batch Studies and Evaluation of Adsorption Ability

Batch adsorption was performed as follows: one-component metal ion solutions of Cd (II) were prepared using cadmium chloride. Samples were measured at different time intervals (0, 5, 10, 30, 60, 120, 240, 480, 720, 960, and 1440 min) to evaluate the effect of contact time. For the effect of adsorbents dosage, 5, 25, 50, 100, and 200 mg of Fe_3_O_4_@PB were added to polypropylene tubes containing 50 mL of 100 μg·L^−1^ Cd (II) solution. Solution pHs were adjusted in the range of 2 to 11 to test the effect of initial pHs. To evaluate the effect of co-existing ions, 5 mg of Fe_3_O_4_@PB was mixed with 50 mL of 100 μg·L^−1^ Cd (II) together with 50, 100, 200, and 400 μg·L^−1^ of Cu (II), Ni (II), and Zn (II) mixture, respectively. Finally, to determine the effect of initial concentration, 5 mg of Fe_3_O_4_-FeHCF was mixed with 50 mL of Cd (II) solutions of 50, 100, 200, 500, 800, and 1000 μg·L^−1^.

All the experiments (unless otherwise noted) were conducted in 50 mL polypropylene tubes and vigorously shaken (150 rpm) at a constant temperature of 25 ± 1 °C for 2 h. The pH of all solutions was set approximately at 6, except for the controlled experiments of pHs. Seven milliliters of all the samples were collected after being filtered using a 0.22 μm mixed cellulose ester membrane, and the concentration of Cd (II) was determined by an inductively coupled plasma mass spectrometer (ICP-MS, Thermo Fisher Scientific ICP Q, Waltham, MA, USA).

### 3.5. Analytical Methods

#### 3.5.1. Calculation of Removal Efficiency and Capacity

The removal efficiency (*R*, *%*) and removal capacity (*q_t_*, μg·L^−1^) of Fe_3_O_4_@PB for Cd (II) were calculated according to Equations (1) and (2):*R* = (*C*_0_ − *C_t_*)/*C*_0_ × 100,(1)
*q_t_* = (*C*_0_ − *C_t_*) × *V*/*m*,(2)
where *C*_0_ and *C_t_* (μg·L^−1^) stand for the concentration of Cd (II) at time 0 and *t*; *V* is the volume of solution in mL, and *m* is the mass of sorbents in g.

#### 3.5.2. Adsorption Kinetics Study

In order to investigate the adsorption kinetics of Cd (II) by Fe_3_O_4_@PB, pseudo-first-order (Equation (3)), pseudo-second-order (Equation (4)), intraparticle diffusion (Equation (5)), and Elovich (Equation (6)) kinetic models were applied:ln (*q_e_* − *q_t_*) = ln *q_e_* − *k*_1_ × *t*,(3)
*t*/*q_t_* = 1/(*k*_2_ ×*q_e_*^2^) + *t*/*q_e_*,(4)
*q_t_* = *K_p_* × *t*^1/2^ + C,(5)
*q_t_* = 1/*β* × [ln (*α* × *β*) + ln *t*],(6)
where *q_e_* and *q_t_* represent the adsorption amounts of Cd (II) at the adsorption equilibrium time and time *t* in μg·g^−1^; and *k*_1_ and *k*_2_ are the rate constants; *K_p_* is the intraparticle diffusion rate constant, and *C* is the intercept. *α* is the initial uptake rate (mg·g^−1^·min^−1^), and *β* is the degree of activation energy and surface coverage (g·mg^−1^).

#### 3.5.3. Adsorption Isotherms Study

In this paper, four adsorption isotherm models were used to express the equilibrium adsorption of Cd (II). Langmuir, Freundlich, Temkin, and Dubinin–Radushkevich isotherms used in this study are shown in Equations (7)–(10) respectively:*C_e_*_/_*q_e_* = 1/(*K_L_* × *q_m_*) + *C_e_*_/_*q_m_*,(7)
ln *q_e_* = ln *K_F_* + ln *C_e/_n*,(8)
*q_e_* = *R* × *T*/*b_T_* × (ln *a_T_* + ln *C_e_*),(9)
ln *q_e_* = ln *q_m_* − *K_DR_* × *ε*^2^,(10)
where *C_e_* is the equilibrium concentration of Cd (II) in the liquid phase in ug·L^−1^; *q_e_* and *q_m_* are the equilibrium uptake amount and maximum monolayer coverage capacity of Cd (II) in ug·g^−1^; *K_L_* and *K_F_* are Langmuir and Freundlich model equilibrium constants, respectively; and *1/n* is the Freundlich index. *R* is the gas constant, 8.314 (J·mol^−1^·K^−1^), *T* (K) is the temperature, *b_T_* (kJ·mol^−1^) is the adsorption heat of Temkin isotherm. *K_DR_* is the activity coefficient of the Dubinin–Radushkevich isotherm, and *ε* is Polanyi potential.

## 4. Conclusions

This work described the removal of Cd (II) from micro-polluted water using novel magnetic core-shell Fe_3_O_4_@PB composites as adsorbents. The adsorption characteristics and mechanisms of Cd (II) on Fe_3_O_4_@PB were studied in detail. The formation of the PB nanoparticles on the Fe_3_O_4_ surface was confirmed by TEM, XRD, and FTIR. Fe_3_O_4_@PB was negatively charged in a wide range of solution pH > 2.31, showing significance for Cd (II) adsorption. A slight acid environment is favorable for the adsorption process, and removal efficiency of 98.78% could be achieved at low initial Cd (II) concentrations, which means a much lower remaining Cd (II) concentration than the DWRLs. The Fe_3_O_4_@PB adsorbents also presented an excellent adsorption efficiency for the removal of heavy metal ions in the presence of several co-existing ions. Results from the adsorption kinetic model imply a chemical-dominated adsorption process. Furthermore, Fe_3_O_4_@PB could be easily separated from the aqueous solution using an external magnetic field. This paper demonstrates that the as-prepared Fe_3_O_4_@PB can be taken as a promising adsorbent for the removal of Cd (II) from micro-polluted water, which could greatly reduce the potential risks of Cd micro-polluted effluent.

## Data Availability

Not applicable.

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
