# Peer review of "Removal of Cd(II) from Micro-Polluted Water by Magnetic Core-Shell Fe3O4@Prussian Blue"

_molecules, 2021, doi:10.3390/molecules26092497_

Round 1

Reviewer 1 Report

The article by Long et al. refers to the removal of Cd(II) ions from water by the magnetic core Fe3O4@Prussian blue. This manuscript is generally of good quality and well organized. However, there are a number of issues that need to be taken into account before its acceptance for publication.

1) The paragraph between lines 343-349 should be deleted because it is information given by Editorial MDPI

2) The last paragraph should summarize the outline of the whole manuscript, however, this seems to be missing.

3) There is a particularly important issue that refers to the 40% plagiarism detected by a professional plagiarism software (without considering references). For example, the paragraph between lines 261-265 should be rewritten because it is exactly the same. There are many more cases I have been able to find. Authors should probably review the current manuscript significantly and solve this problem very carefully. In addition, once the author has finished writing, authors must perform a plagiarism evaluation using professional plagiarism software.

Reviewer 2 Report

Removal of heavy metal from aqueous solution is important topic in environmental issue. Many studies reported the new materials for removal of heavy metal. 

The paper entitled "Removal of Cd(II) from micro-polluted water by magnetic core-shell Fe3O4@Prussian blue" is interesting that contains some new findings.

However, the authors have to address the following points before publication in the journal Materials:

  1. Abstract should add detailed characterization of core-shell material.
  2. The Introduction included the comprehensive literature review on the toxicity and the occurrence of heavy metal Cd.
    Different techniques for removal of Cd were also introduced in details. But some novel adsorbents with surface modification should be emphasized.
    The authors needs refer some papers.
    Journal of Molecular Liquids 287, 110900 (2019); Langmuir 2021, 37, 9, 2963–2973; Journal of Molecular Liquids 309, 113150 (2020); Journal of Molecular Liquids 301, 112456 (2020).
  3. Introduction should avoid "the described as follows".
    The novelty should emphasize the novelty of the present study. 
  4. The pore structure of Fe3O4@PB should be confirmed.
  5. The resolution of Figure 2 is too low. 
  6. The zeta potential of the materials results should be discussed in more details.
  7. Figure 4 and caption should be in one page.
  8. Many interferences affected to adsorption. Why did the author study Cu, Ni, Zn.
  9. The R2 of Langmuir model is wrong. Please check this value
  10. Conclusions should again emphasize the new quantitative findings in the present study.

Round 2

Reviewer 2 Report

The revised paper is significantly improved by the authors.

It can be published in the current form.